# The Versatility of HVOF Burner Rig Testing for Ceramic Matrix Composite Evaluation

**Gregory N. Morscher \*, Ragav P. Panakarajupally and Leland Hoffman**

Department of Mechanical Engineering, University of Akron, Akron, OH 44325, USA; rp95@uakron.edu (R.P.P.); lch32@uakron.edu (L.H.)
\* Correspondence: gm33@uakron.edu; Tel.: +1-330-972-7741

**Abstract:** Effective testing of ceramic matrix composites (CMCs) and CMC/coating systems for high temperature, high stress, high velocity and/or severe oxidation/corrosion environments is a critical need in materials/coatings evaluation for extreme environments of hot section parts in jet engine and hypersonic applications. Most current technology can evaluate two or three of the extreme conditions for a given application; however, incorporating as many of the extreme thermo-mechanical-environmental factors is highly advantageous to understand combinatorial effects. A high velocity oxygen fuel (HVOF) burner rig offers an excellent platform to evaluate many of these extreme conditions. In this work, the following three different thermo-mechanical-environmental test conditions using an HVOF rig on SiC-based CMCs are highlighted: (1) fatigue at temperature for >Mach 1 velocity and high temperature compared to typical stagnant air test environment, (2) high temperature hard particle erosion at temperature for ≤Mach 1 conditions and (3) ~Mach 5 near-hypersonic velocity conditions at very high temperature exposure.

**Keywords:** ceramic matrix composite; burner rig; high temperature testing

## 1. Introduction

Ceramic matrix composites have recently been introduced into commercial jet engine hot-section parts [1]. These necessitate some of the following most extreme conditions for a material: high temperature, relatively high stress (including those induced by multi-directional thermal gradients), oxidative environments, corrosive environments, high velocity and/or long times. Other less mainline applications, such as hypersonic components, look forward to even higher stresses, temperatures, extreme environments and/or velocities. Adequate testing for these types of conditions is a challenge in and of itself. Current test facilities [2–20] consist of combustion, arc-plasma, laser or combinations thereof to generate the temperatures, velocities and/or environments of interest (Table 1). Most of these rigs can effectively evaluate two or, at most, three of the above conditions. Apart from the AFIT HVOF rig in Table 1, applied stress is not incorporated into the rig; however, for many of the rigs, sufficiently large specimens can be inserted that can later be tested for retained mechanical properties.

Increased opportunities and the use of these types of composites are expected to expand dramatically and better test evaluation approaches are needed to assess the combinatorial impacts of the different thermal, stress and environmental conditions to determine the effectiveness of ceramic matrix composite systems, which may include protective coatings, for such applications. This necessitates that pertinent test rigs need to be simple, cost effective and yet quantitative as to the thermal, stress and environment conditions imposed on the test specimen. Related to this is the need for diagnostic techniques to measure thermal gradients and local strain as well as health-monitoring techniques to assess the progressive damage that occurs during the test.

**Table 1.** Some elevated temperature/velocity test rigs for jet engine and hypersonic materials testing.

| Facility | Type | Maximum Velocity (Mach) | Maximum Temp. (°C) |
|---|---|---|---|
| NASA Mach 0.3 BR [2] | Jet fuel combustion | 0.3 | 1316 |
| NASA High Pressure (4–15 atm) [3] | Jet fuel combustion | 0.1 | 1550 |
| AFIT HVOF [4] | Combustion | 1 | 2500 |
| HYMETS [5–7] | Arc-plasma | 5 | 2480 |
| NASA Combustion Scramjet * [8] | Combustion Scramjet | 6 | 1400 |
| NASA Arc-Heated Scramjet * [9–11] | Arc-heated Scramjet | 8 | 2600 |
| GHIBLI [12] | Arc-plasma | 10 | 9700 |
| SCIROCCO * [13–16] | Arc-plasma | 12 | 9700 |
| Pprime MAATRE * [17] | Burner Rig | 1 | 1600 |
| VKI Plasmatron * [18] and other plasma [19–21] | Induction-Plasma | 1 | 11,700 |
| DRL HEG* [22,23] | Free-Piston Shock Tunnel | 10 | 1176 |

* Capable of large specimen size (>250 mm).

Combustion test rigs, such as high velocity oxygen fuel (HVOF) burners, are ideal sources for elevated temperature (>2000 °C including axial and through-thickness thermal gradients), high velocity (up to ~950 m/s or 1700 m/s depending on the nozzle) and relatively high water contents (up to ~0.4 atm depending on combustion condition) more in line with engine applications [24–26]. An HVOF designed for particle spray deposition also has the added benefit of incorporating a particle feeder that enables simulation of high melting point hard particle erosion [27] or lower melting point sand ingestion, which results in molten glass deposition on the surface of the material, i.e., CMAS (calcium-alumina-magnesium-silicate). In this study, we highlight the use of an HVOF burner rig with the capability of uniaxial load conditions to apply stress-cycle and/or thermal/velocity conditions at temperature. The focus in this work will be the burner rig exposure of CMCs under a variety of stress/temperature/environmental conditions; however, the facility is ideal for CMC/EBC or metal/TBC systems as well. Some of what has been presented here has been published in earlier works [25–27] that will be highlighted and discussed as well as newer work aimed at elevated temperature, higher velocity conditions.

## 2. Experimental

### 2.1. Burner Rig Facility

The burner rig facility is shown in Figure 1. A high-velocity oxygen fuel (HVOF) gun (HP 2700 HVOF system, Plasma Powders & Systems, Marlboro, NJ, USA) is used to simulate the combustion environment. Two different nozzles are employed, one that operates at lower velocities and another that can operate at higher velocities. For the lower velocity nozzle, the HVOF system uses propane as fuel, oxygen as oxidizer and compressed air to cool the nozzle. A peak temperature of 2200 °C is possible with expanding gas velocities approaching 943 m/s, which can be achieved based on the equivalence ratio (ratio of actual fuel to air to stoichiometric fuel to air) of gases. The flame diameter at the nozzle exit is 3 or 4 mm. The diameter of the hottest section of the flame is about 10–15 mm as the flame expands away from the nozzle resulting in axial thermal gradients along the length of a specimen. Lower temperatures are easily achieved by moving the nozzle away from the specimen. A horizontal hydraulic MTS machine (MTS 810 system, Eden Praire, MN, USA) was incorporated to simulate mechanical loading. Different combustion conditions such as temperature, % water vapor and velocity can be achieved by adjusting the flowrates and pressures of propane and oxygen. This facility can simulate both fuel rich and fuel lean conditions. Test coupons can be mounted in the hydraulic grips at various angles, ~20 to 90° of the flame produced by the HVOF gun. Heating of the front surface of the specimen and the natural convection that occurs on the back surface creates thermal gradient stresses. Steel plates and multilayer ceramic insulation were used to protect the grips and wedges from excessive heating. For the higher velocity nozzle, water cooling is required and velocities approaching Mach 5 (hypersonic conditions) can be obtained. The

maximum velocity achieved to date using the water-cooled nozzle is ~1700 m/s (Mach 4.9) (Improvements are currently underway that should enable velocities approaching Mach 7).

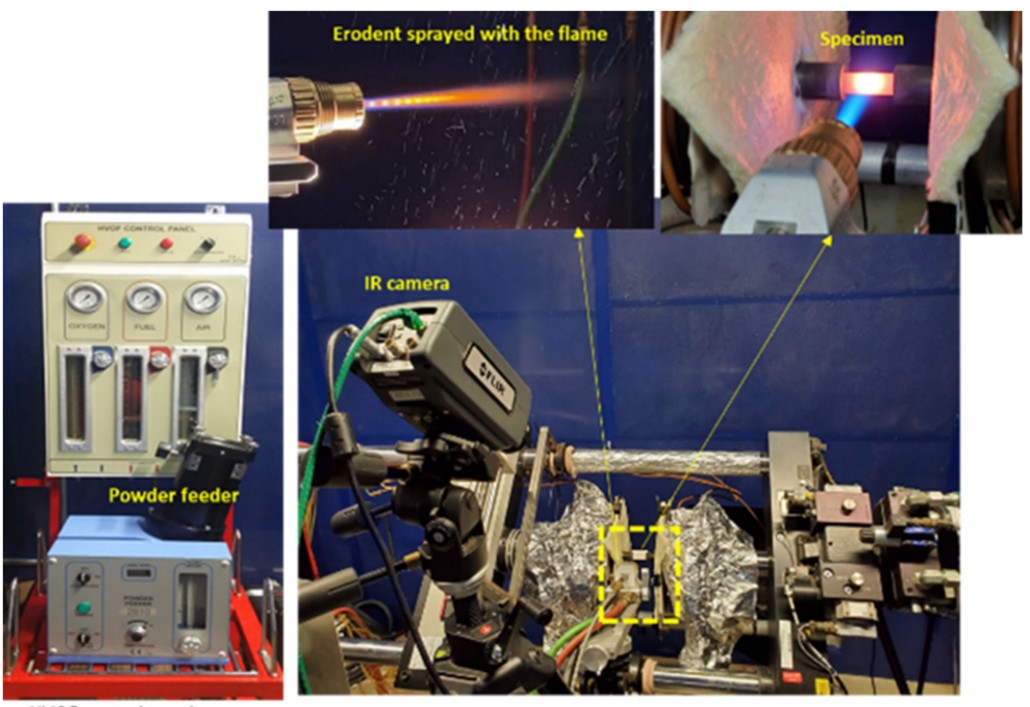

**Figure 1.** University of Akron HVOF Burner Rig.

To simulate erosion conditions, the control panel consists of a powder feeder where powders can be ingested. The surface temperature condition was established and held for three minutes to stabilize the specimen temperature. The particles (0.2 g) were fed to the flame using nitrogen as the carrier gas. The feed rate of the erodent can be controlled by adjusting the rpm of the disc (located inside the feeder) and flowrate of nitrogen gas. The specimen was then cooled down and the erosion crater size and mass were measured. This step was again repeated for a total of five iterations. For the work presented here, three different particle size alumina powders were used as the erodent materials.

Two forward looking infrared cameras (FLIR A6700SC, FLIR Systems, Wilsonville, OR, USA) were used to monitor the front and back surface temperature of the specimen by inputting the known emissivity of the specimen. Emissivity was calculated using an AMTECO Inc. (Model HRFS-275-2Z, West Chester, OH, USA) resistive furnace with a sapphire window opening to monitor the front surface temperature of the specimen using an FLIR camera. Temperature of the furnace was raised to the set temperature (measured with R-type thermocouple using AMTECO controller) or highest temperature of the furnace (1315 °C) for higher temperature burner rig conditions with a representative specimen. Emissivity in the FLIR camera was adjusted until the furnace and the FLIR temperatures matched.

Resistance was measured via the four-point probe technique in some tests to monitor changes in resistance that could be correlated with temperature changes and/or damage development. The current and potential drop leads were placed at the ends of the specimen, as shown in Figure 2, in the region of either the grips or mounting fixture (if no load is applied), which is well insulated and far from the flame.

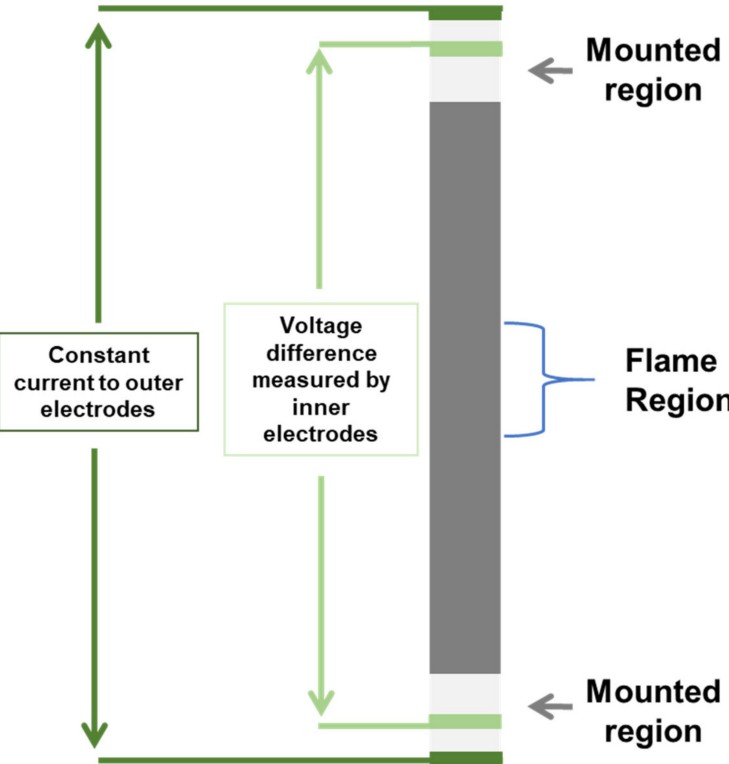

**Figure 2.** Schematic of electrical resistance measurement.

*2.2. Experiments and Materials*

Three different cases (Table 2) are presented here that involved available SiC-containing CMCs. Table 2 describes the tests performed and composite systems examined. Table 3 describes the HVOF test conditions for each case. Based on the theoretical equations for combustion, the maximum reactive species in the flame for Case 1 would contain ~11.9% $H_2O$ and 17.7% $O_2$, whereas Case 3 (the other extreme) would contain ~7.3% $H_2O$ and ~20.6% $O_2$. For Cases 1 and 2, the nozzle tip was about 125 to 150 mm from the specimen surface, whereas for Case 3, the nozzle tip was about 60 to 75 mm from the specimen surface.

**Table 2.** Case Studies and Composite Materials.

| Case | Burner Rig Experiments | Materials [a] |
|---|---|---|
| 1. Effect of Burner Rig compared to Typical Furnace Environment | Fatigue at 1200 °C while subject to burner rig (650 m/s) or standard resistance heated furnace.<br>Fatigue conditions: R = 0.1; 1 hz<br>Angle of flame incidence = 45° | Hi-Nicalon Type S (Nippon Carbon Inc., Japan) referred to as HNS and Tyranno SA (Ube Industries, Japan) woven fiber reinforced melt-infiltrated composites fabricated by the former Goodrich Corporation similar to those in Reference [28] with a 0.5-micrometer BN interphase. Both are 8 ply five-harness satin. HNS thickness was 2.7 mm with a fiber volume fraction of 0.3, whereas SA thickness was 2.0 mm with a fiber volume fraction of 0.36. |
| 2. Use of HVOF Burner Rig to Understand High Temperature Particle Erosion | Hard particle (alumina) erosion for three different particle sizes, surface temperatures at either 815 or 1200 °C and two different velocities (200 and 350 m/s). The largest particle size erodent was tested at three different incidence angles (30, 60 and 90°). | Tyranno SA (Ube Industries, Japan) 2D woven (five-harness satin; 8 ply) fiber reinforced melt-infiltrated composites same as Case 1 (similar to [28]) |

**Table 2.** *Cont.*

| Case | Burner Rig Experiments | Materials [a] |
|---|---|---|
| 3. Use of HVOF Towards Hypersonic Conditions | Five iterations of short time (2 min) exposure of SiC-based composites to 1700 m/s and surface temperature of 1650 °C followed by cool down to room temperature. No load was applied for these experiments. Electrical resistance was monitored (130 mm inner lead distance) to understand the response of the composite to the temperature/velocity condition [b]. Angle of flame incidence = 90° | The following three SiC-based woven composites were tested: (1) 2D woven (five-harness satin) HNS reinforced, BN interphase, full CVI SiC matrix composite (152 mm × 12.7mm × 2.3mm, 0.33 fiber volume fraction) [24]; (2) SA-Tyrannohex (152 mm × 10 mm × 2 mm) [29], a hot-pressed woven SA fiber-only SiC material; and (3) Tyranno SA (152 mm × 10 mm × 2 mm) (Ube Industries, Japan) woven fiber reinforced melt-infiltrated composite the same as Case 1 (similar to [28]) |

[a] All specimens had as-machined edges, which is expected to worsen mechanical degradation compared to sealed edges. [b] Electrical resistance was also monitored in the previous two cases, but the results are not presented here as the focus is more on the use of HVOF. The details pertaining to the change in resistance with thermo-mechanical exposure for the first two cases can be found in the appropriate references.

**Table 3.** Case Studies and HVOF Conditions.

| Case | Particle Size (μm) | Gas | Pressure (Psi) | Flow Rate (slpm) | Velocity (m/s) |
|---|---|---|---|---|---|
| 1. Effect of Burner Rig compared to Typical Furnace Environment | | Propane | 80 | 50 | |
| | | Oxygen | 150 | 280 | 650 |
| | | Compressed Air | 75 | 280 | |
| 2. Use of HVOF Burner Rig to Understand High Temperature Particle Erosion | 125 | | 50, 100, 75 | 22, 110, 150 | 200 |
| | 125 | | 50, 100, 70 | 30, 150, 200 | 350 |
| | 210 | Propane, Oxygen, Compressed Air | 65, 125, 70 | 30, 150, 150 | 200 |
| | 210 | | 65, 125, 70 | 45, 250, 300 | 350 |
| | 305 | | 80, 140, 70 | 30, 200, 200 | 200 |
| | 305 | | 80, 140, 70 | 55, 330, 400 | 350 |
| 3. Use of HVOF Towards Hypersonic Conditions | | Propane | 110 | 60 | |
| | | Oxygen | 160 | 400 | 1700 |
| | | Compressed Air | 75 | 580 | |

For Cases 1 and 3, velocity was calculated by spraying a small amount of chromium carbide particles, initially at the HVOF conditions for a given experiment, via the powder feeder (Chrome carbide particles glow bright when heated and passed through the flame, which is easily observed with the high speed camera). A high-speed camera was used to record the images of the particles sprayed. For the erosion tests of Case 2, specific erodent particles could be used for velocity measurements. Velocity was calculated by tracking the distance travelled by the particles at a given time using an open-source software called "Tracker" ("Tracker Video Analysis and Modeling Tool for Physics Education." https://tracker.physlets.org/; accessed on 19 August 2021).

## 3. Results and Discussion

Three cases are highlighted in this work for the testing of SiC-based composites using the HVOF environment. More quantitative studies on more mature and advanced composite systems need to be performed but the purpose of this work was to demonstrate some of the differences when testing using the combustion flame HVOF environment compared to conventional elevated temperature testing utilizing stagnant air furnace conditions and some capabilities of the HVOF test approach. The composites used in these studies are those that were available to the authors and do not represent the most mature composite systems currently available. However, they are believed to show, to a degree, the general responses of SiC-based composites to these types of environments. Note that the first two cases are taken from studies recently published and more detail is given in the

referenced works. The third case is unpublished research and represents early findings on the use of HVOF burners for near-hypersonic conditions with these types of materials.

### 3.1. Effect of Burner Rig Compared to Typical Furnace Environment under Fatigue

Historically, most elevated temperatures under stress conditions have been performed using furnaces where there is an axial temperature gradient but relatively uniform temperature "hot zone" of some length. The high velocity burner rig environment subjects the specimen to not only an axial thermal gradient, but also a significant through-thickness thermal gradient. Figure 3a shows the more severe degradation of the burner rig environment compared to the resistance heated furnace environment subject to fatigue (1 Hz, R = 0.1) for a max temperature of 1200 °C for an HNS and SA MI woven composites [25,26]. The burner rig environment resulted in at least an order of magnitude shorter lifetimes compared to the standard furnace environment. The front and back side temperature profiles of the burner rig tests for the two composite systems are also shown (Figure 3b,c). The through-thickness thermal gradient for HNS was about 175 °C, whereas for the SA composite it was about 100 °C. The difference in thermal gradients was most likely due to the different thicknesses of the specimens: 2.7 mm for HNS and 2.0 mm for SA.

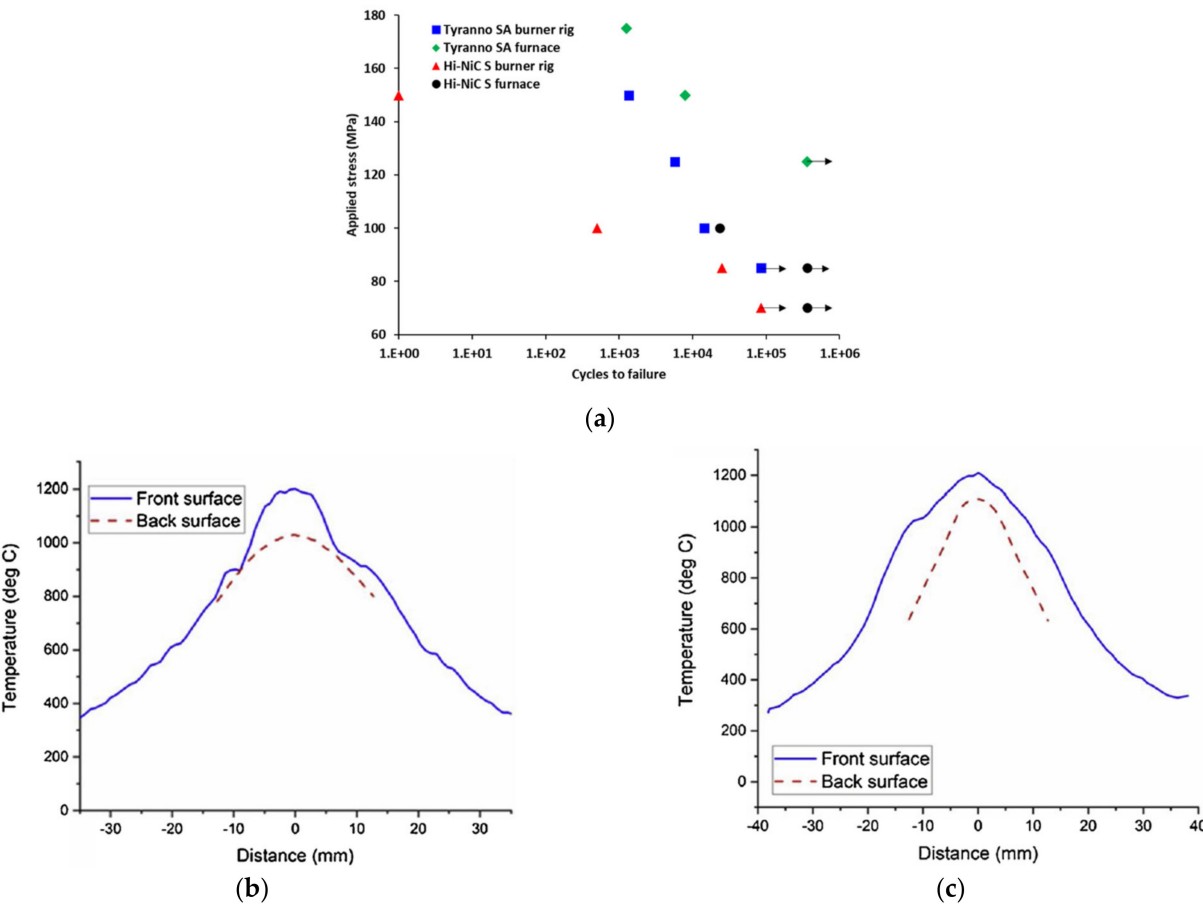

**Figure 3.** (**a**) Stress vs. cycles to failure for HNS and SA MI composites subject to burner rig and resistance heated furnace environment with maximum temperature of 1200 °C. The burner rig exposure resulted in a through thickness thermal gradients for (**b**) HNS MI and (**c**) SA MI, respectively. (After [25,26]).

The severity of the burner rig condition demonstrates the importance of considering the thermal gradient and velocity for materials behavior. The resistance heated specimen had an axial hot zone length of 25 mm, whereas the burner rig specimen had a hot zone length of about 10–12 mm, which was at its highest temperature only at the surface of the specimen. It is conservatively estimated that the volume of material subjected to the highest

temperature (~1200 °C) was at least four or five times greater for the furnace condition compared to the burner rig condition. Therefore, the volume exposed to temperature is not necessarily the most critical issue. It appears that the thermal gradient, velocity conditions and increased H2O content of the environment for the burner rig test enhances the oxidation embrittlement mechanisms leading to failure and/or introduce new damage mechanisms that reduce life. Two features were observed to this effect as follows.

The typical phenomena of stressed-oxidation embrittlement [30,31] where oxidation through surface cracks results in the internal oxidation of interphases, fibers and matrix causing the strong bonding of fibers to the matrix was evident in the fracture surfaces for both the furnace and burner rig tested specimens from fracture surface analysis. However, one feature that was unique for many of the SA burner rig specimens was that oxidation embrittlement was more prevalent from the backside of the specimen compared to the front side (Figure 4). It is presumed this was due to the thermal gradient. If the front side is at a higher temperature than the back side under load, then the front side will relieve some stress due to higher thermal strain that must be compensated for on the back side, which should result in larger crack openings enabling greater environmental ingress into the specimen.

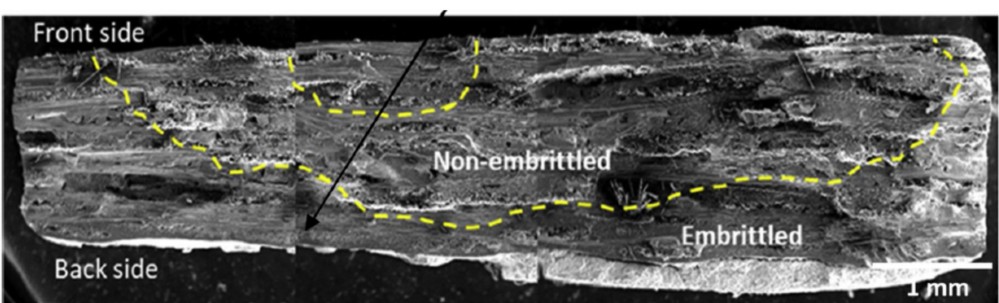

**Figure 4.** SEM fracture surface of SA-MI 150 MPa burner rig test.

The second feature observed for both composite types under burner rig and stress exposure was the presence of longitudinal crack formation (Figure 5). This must be due to the more complex stress state imposed by the applied stress and thermal gradient. There were no longitudinal cracks observed in any of the furnace-tested specimens. The enhanced damage would interact with more typical transverse cracks to increase pathways for oxidation ingress as well as reduced local modulus. There are significant large pores in the HNS MI panel composite (Figure 5a), which is expected to increase the ingress of oxidizing species and increase the propensity for interlaminar cracking. Nevertheless, interlaminar cracking was never observed for the furnace-tested specimens and occurred in both the HNS and SA composites.

In addition, this type of testing presents an excellent approach to evaluate environmental barrier coated (EBC) specimens, with stress, since a typical engine application will involve a CMC-EBC combination. Some work to this end was performed with slurry-derived EBC coatings and it was evident that the coated specimens outperformed the uncoated substrate composite materials [24].

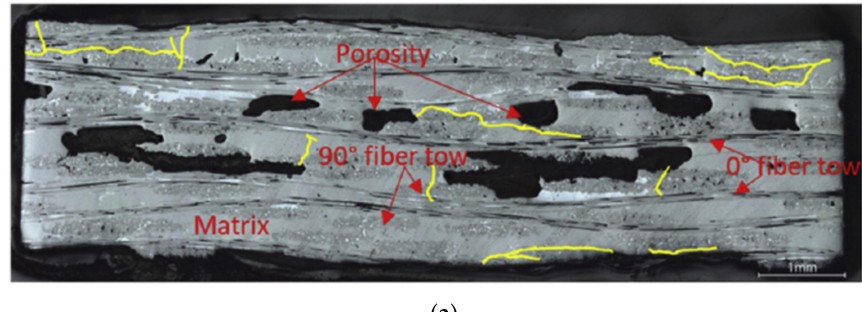

(**a**)

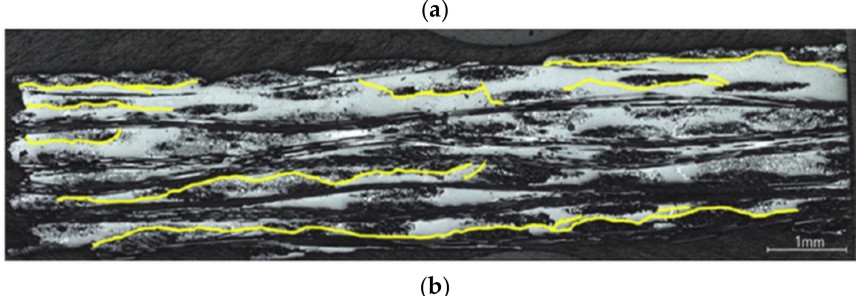

(**b**)

**Figure 5.** Optical micrograph of polished longitudinal section (stress in the horizontal direction) for burner-rig tested (**a**) HNS at 100 MPa and (**b**) SA at 125 MPa. Cracks are highlighted in yellow. (From [23]).

### 3.2. Use of HVOF Burner Rig to Understand High Temperature Particle Erosion

Many jet engine application environments will include particle ingestion into the engine whether from a tarmac, desert or volcanic ash. Such conditions can result in hard particle erosion or molten deposition onto CMC and/or CMC/EBC materials. For hard particle erosion, considerable work has been performed by Choi and coworkers on many different substrate materials at room temperature [32,33]. However, no work had been performed at an elevated temperature. As an initial study, Panakarajupally et al. [27] used the HVOF burner rig that is designed for particle spraying of elevated temperature coatings to study the effect of hard particle erosion on the SA-MI composite similar to that used in Case one. Alumina was used as the erodent because it was found to maintain particle structure and shape as it passes through the flame due to its relatively high melting point (2050 °C). Garnet, which has been used in many of the previous room temperature studies [29,30], was found to melt to a greater degree when passed through the HVOF flame due to its lower melting point (~1930 °C). The specimen size for the erosion tests was 38 mm × 12.5 mm and a specimen thickness of ~2.3 mm.

Three different particle sizes at two different elevated temperature (815 and 1200 °C) conditions and two different velocities (200 and 350 m/s) were employed. Note that the temperature condition is that of the surface temperature of the specimen as controlled by the distance of the specimen surface from the nozzle tip. The particles themselves go through the same combustion environment of the HVOF nozzle and cool down with the flame. The particle temperature is expected to be slightly higher than the flame temperature upon impact. Most of the tests were performed with an impingement angle of 90° (normal) to the surface; however, for the smaller mesh (larger particle size powder), tests were also performed at 30 and 60° impingement angles to discern the effect of the impingement angle on erosion.

The effect of velocity and temperature on the erosion rate is shown in Figure 6. Note that erosion rate is defined by mass loss as follows:

$$\text{Erosion rate} = (m_i - m_f)/m_e \tag{1}$$

where $m_i$ is the initial mass, $m_f$ is the final mass after erosion and $m_e$ is the mass of the erodent. It has been reported in the literature [34,35] that the erosion rate is proportional to

the velocity according to a power law relationship where "n" is the exponent of the power law. At room temperature, the n for SiC-based ceramics has been found to be ~2, which corresponds to brittle fracture mechanisms [33–35]. It was found in [27] that n was two at room temperature for these SA-MI composites.

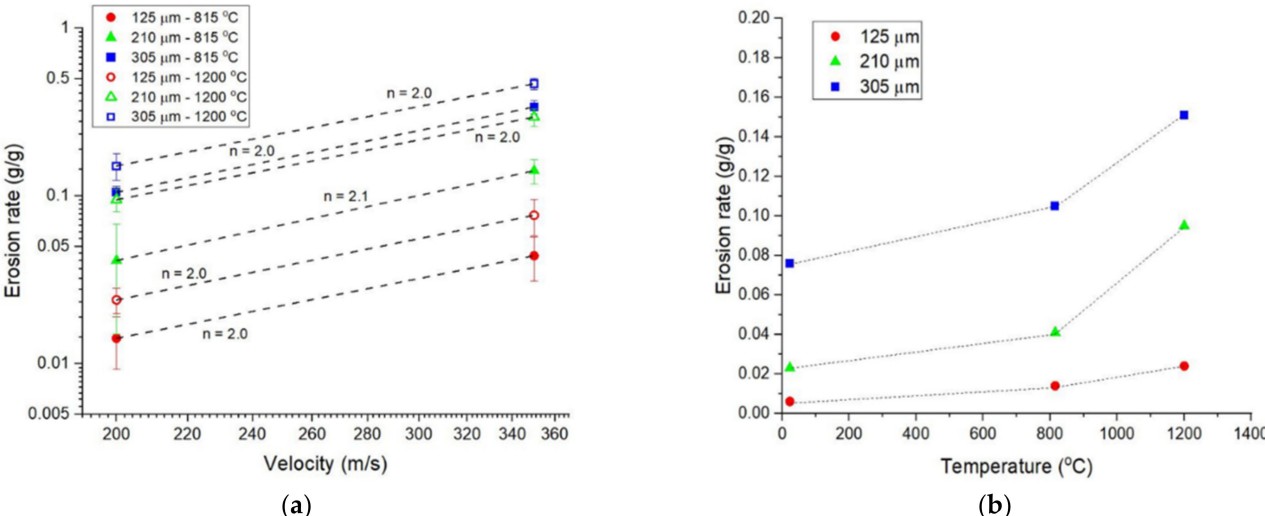

**Figure 6.** Erosion rate as a function of (**a**) velocity at 815 and 1200 °C under normal incidence impact conditions and (**b**) temperature for three particle sizes under normal incidence impact conditions at a velocity of 200 m/s. After [27].

It is evident that particle size, velocity and temperature all have a strong effect on the erosion rate with the velocity exponent having a value of two, the same as the room temperature experiments (Figure 6a). The largest particle size erodent was the most erosive for both temperature and velocity conditions.

Figure 6b shows the effect of temperature on the erosion rate for the velocity condition of 200 m/s. It is evident that the erosion rate increases with temperature. For all particle sizes, the erosion rate was more than a factor of two at 1200 °C compared to room temperature.

Figure 7 shows the effects of the impingement angle and erodent concentration at 1200 °C on erosion for the largest particle size powder (305 micron) at 350 m/s. Erosion is greatest at an impingement angle normal to the specimen surface and decreases as the angle is decreased (Figure 7a). The "predicted" line shown in Figure 7a is the prediction assuming elastic fracture mechanisms controlling the erosion of the normal incidence condition (n = 2). The fact that the erosion rate at lower angles of incidence is higher than the prediction implies "ductile" or ploughing mechanisms that enhance the removal of material. When the erodent concentration is increased per erodent iteration (the baseline used was 0.2 g), the erosion rate decreases, though erosion itself may increase. The decrease in the erosion rate is most likely due to the increased interaction of erodent particles where particle-to-particle collisions occur prior to impacting the substrate material, thereby lessening or negating the contribution of some of the particles to erosion. These data demonstrate that for modeling erosion for CMCs, it is important to understand the concentration of erodent particles for jet engine applications when using composites.

The HVOF rig with particle ingestion capability can be quite useful for understanding various ingestion phenomena pertaining to different environments for turbine or other air-breathing elevated temperature engine applications—a phenomena not well understood for many materials. Though the HVOF combustion conditions are much higher than typical jet engine max temperatures, the phenomena of either non-melting particles, partially melting or fully melting particles can be attained with the proper composition of particles used and augmented to simulate conditions similar to those in an engine. Future work in this

area should include the ingestion of lower melting point sand particles to simulate the deposition of molten CMAS compositions on CMC and CMC/EBC substrates.

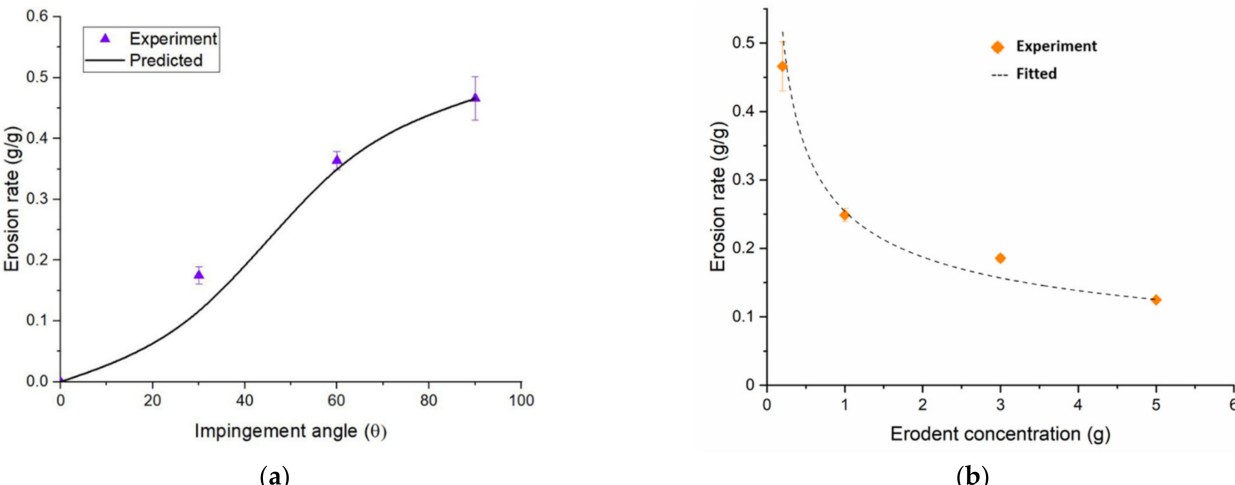

**Figure 7.** Variation of erosion rate for 54 mesh (305 micron) particle size at 1200 °C and at a velocity of 350 m/s for (**a**) impingement angle and (**b**) erodent concentration (error bars refer to standard deviation). After [27].

### 3.3. Use of HVOF Burner Rig towards Hypersonic Conditions

With the proper nozzle, velocities about Mach 5 (1750 m/s) were approached using the HVOF burner rig. To date, we have achieved velocities of 1700 m/s. As a preliminary investigation, three different SiC-based composites readily available to the authors were subject to the same condition: five two-minute intervals of 1700 m/s, 1650 °C surface temperature. Between each iteration, the specimen was allowed to cool to room temperature. The electrical resistance was monitored during the entire thermo-cycle sequence and the mass was measured prior to and after the entire thermo-cycle sequence.

Figure 8 shows some of the details for a burner rig thermo-cycle test for HNS-CVI SiC. The thermal profiles from the FLIR cameras on the front and back sides (Figure 8c,d) were used to determine the max temperature for each side, which are plotted in Figure 8e for the entire test, including cooling. The through-thickness thermal gradient was ~220 °C. Note that the resistance decreases sharply with the sharp increase in temperature demonstrating how sensitive resistivity of SiC is with temperature and that resistance is a good monitor of the test conditions themselves. Upon cooling (turning off the flame), the specimen cools to room temperature and the resistance increases to a value less than what its original pristine value. This is probably because the SiC composites are fabricated at temperatures lower than the thermal exposure and the effect of the higher temperature results in some modification of the microstructure and/or diffusion of atomic species that makes the material slightly more conductive. Additionally, note that upon subsequent thermo-cycle iterations, the residual resistance (Figure 8f) increases slightly with each cycle, especially the latter two cycles. This may be indicative of material removal or damage due to HVOF exposure.

Figure 9 shows optical photographs of the front side, back side and edge of the HNS-CVI composite after the five-cycle thermo-cycle iteration. There is some oxidation and material removal on the front face of the specimen (Figure 9a). The oxidation due to the higher temperature condition in the center of the flame on the back side is also obvious from the discoloration of the specimen (Figure 9b). Interestingly, it was observed that the specimen itself "bent" due to the force and temperature of the HVOF flame. There was a 2° bow in the specimen (Figure 9c) at the center where the flame impinged on the specimen due to the applied bending load from the flame and creep and/or microcracking of the specimen. This is of some concern and may require some backside support or thicker specimens.

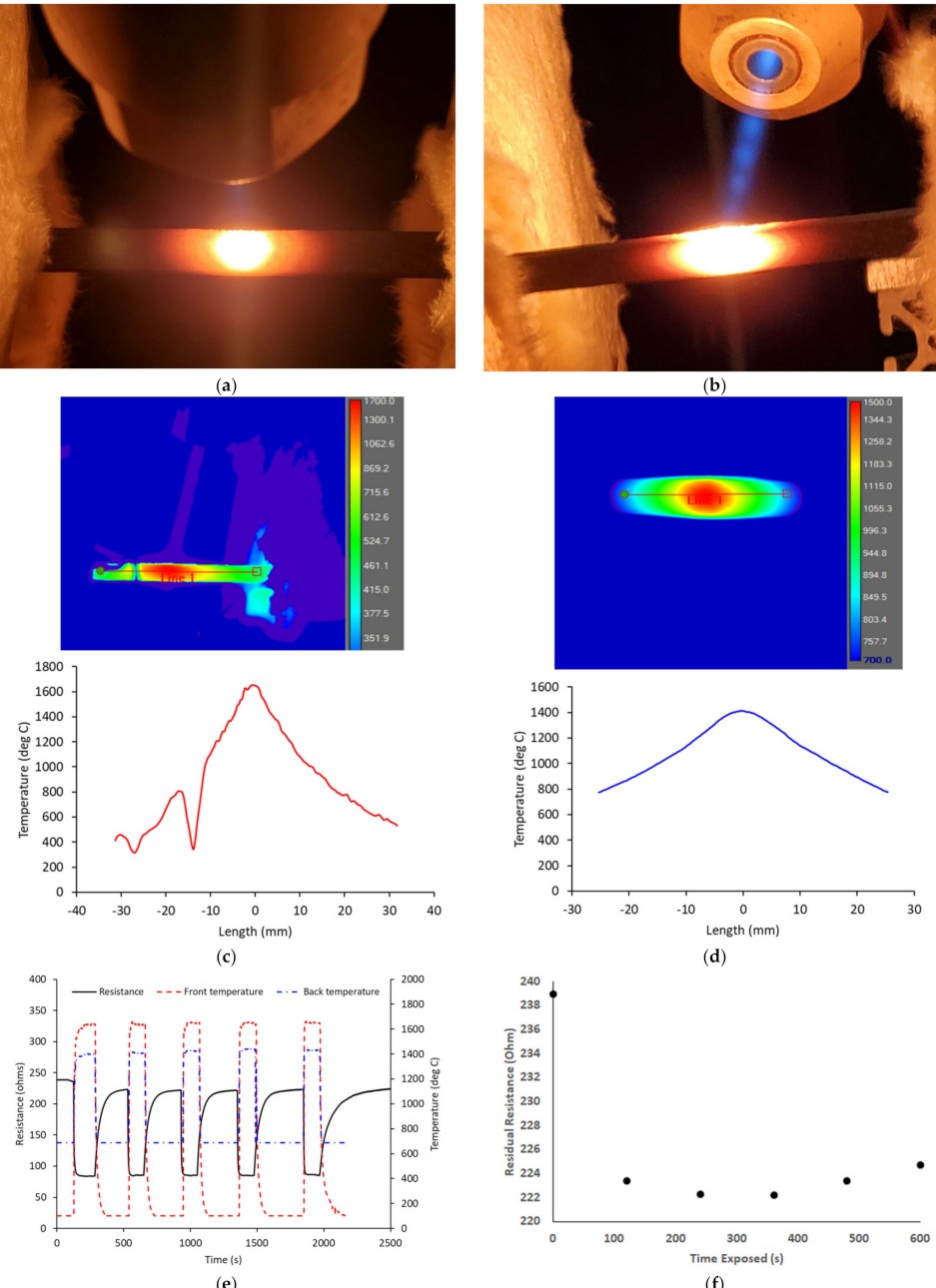

**Figure 8.** Collection of representative features of 1650oC surface temperature, 1700 m/s HVOF burner rig exposure: (**a**) front side image, (**b**) back side image showing HVOF nozzle and blue flame, (**c**) front side thermal image and line scan for axial thermal profile, (**d**) back side thermal image and line scan for axial thermal profile, (**e**) experimental data acquired throughout the test including front side and back side temperature as well as electrical resistance measured from the cold ends of the six inch specimen (inner four-point probe lead length = 130 mm), and (**f**) room temperature measured electrical resistance before and after each thermo-cycle iteration.

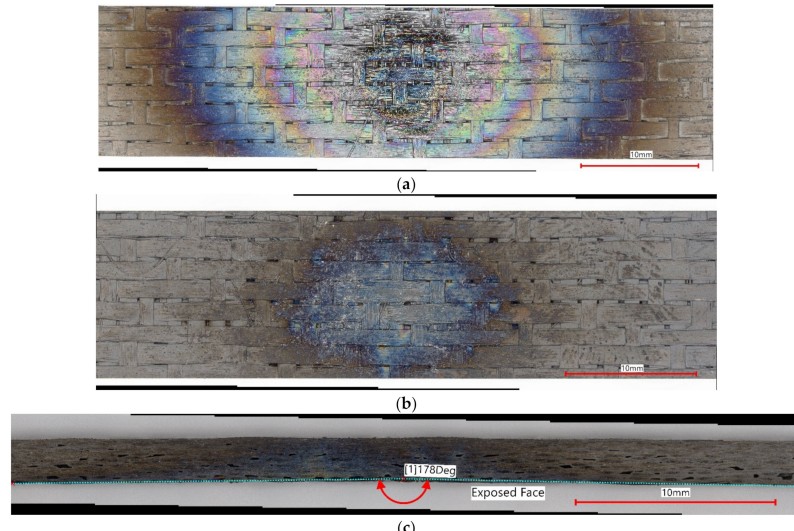

**Figure 9.** Optical micrographs of (**a**) front side, (**b**) back side and (**c**) edge of HNS CVI SiC after HVOF exposure.

The HNS CVI specimen was tested in tension at room temperature to failure and exhibited an elastic modulus of 180 MPa and a strength of 163 MPa. The pristine specimens tested in [25] exhibited an elastic modulus of 200 GPa and a strength of at least 258 MPa. The fractured pieces of the HNS CVI specimen after room temperature retained strength testing showed considerable fiber pullout (Figure 10a). Pullout was especially prevalent on one edge (Figure 10b). Long fiber pullout requires very low interfacial shear strength. Evidently, the short time exposure was sufficient to oxidize the exposed interphases via the exposed edges in order to degrade the BN/fiber interface. Fiber/interface decoupling would account for the observed increase in resistance, reduced E and degraded strength (at least in part). It is likely that the presence of carbon at the surface of the HNS fibers [36] only exacerbates the fiber/matrix interface decoupling.

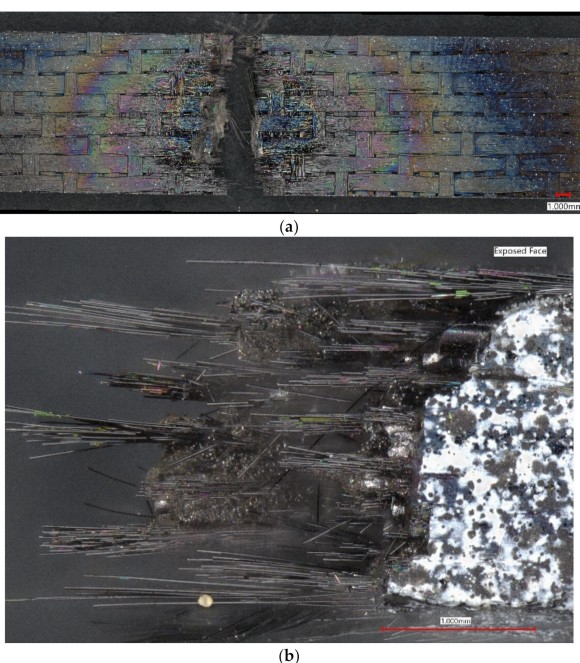

**Figure 10.** Optical micrographs of (**a**) both sides of specimen from the flame-impinged face and (**b**) edge of the specimen showing significant fiber pull-out. Note the speckle pattern in (**b**) was used for strain measure (DIC) during room temperature tensile test.

The SA-Tyrannohex showed quite a different electrical response when subject to the same thermo-cycle condition (Figure 11a). The through-thickness thermal gradient for the SA-Tyranno-Hex material was slightly higher, ~260 °C, than that of the HNS-CVI. The resistance showed an increase and then decrease followed by a more moderate increase in resistance during the two-minute exposure. Upon removal of the flame, a sharp increase (spike) in resistance was observed after each iteration. The data are magnified in Figure 11b to show the increase then decrease in resistance upon initial heating. Note that the decrease is while the temperature is increasing. The dramatic increase in resistance with the removal of the flame is also apparent, followed by a decreasing resistance with the temperature decrease. There are two explanations for the increasing resistance with an increasing temperature up to about 1200 °C. This could be due to the influence of the Al dopant concentration such as the temperature dependent dopant effects observed in MI composites consisting of doped Si in the matrix [26,37]. It is also possible that some of the hot-pressed fibers decouple from one another and alter the circuit and current flow. Nevertheless, the thermal-electrical response was reproducible for multiple cycles and two different specimens. Figure 11c shows a progressive increase in residual room temperature resistance after each cycle. The data are plotted as resistance change as follows:

$$\Delta R = R_{cycle} - R_o \tag{2}$$

where $R_{cycle}$ is the room temperature resistance after a given thermal cycle and $R_o$ is the resistance of the pristine material prior to testing. There appears to be a near linear increase in the increasing resistance change that must be due to the oxidation/damage effects from thermal exposure. There was obvious oxidation (Figure 11d) where the flame impinged on the specimen. There was minimal change in thickness in this region. We were not able to effectively test the straight-sided specimen in tension, it failed in the grip upon initial loading. However, a four-point bending resulted in a strength of 143 MPa, about half that published in the literature (The same inner and outer spans were used as in [38] (20 and 40 mm, respectively); however, the pristine data in the [29] were for a 4-m wide specimen compared to our 10-m wide specimen. The inner four-point probe length was 130 mm) [38].

Note that the changes in resistance for the Tyranno-Hex are slight compared to the HNS-CVI of Figure 8e. Two different specimens were tested of the SA Tyranno-Hex and both showed the same behavior. Since Tyranno-Hex consists essentially of SA fibers alone, the changes in resistance are related to the electrical properties of the fibers themselves and perhaps to the interconnected network of 0/90 fiber linkages formed during the hot-pressing process. The HNS-CVI composites have two major current carriers, the HNS fibers and the CVI SiC. Certainly, oxidation of the interphase, carbonaceous species within the HNS fibers, the temperature dependence of the constituents themselves and damage due to matrix cracking will all complicate the temperature dependent electrical properties of the composite when compared to the Tyranno-Hex material.

The final test was performed on a SA-MI composite (Figure 12). Not surprisingly, the specimen exhibited the worst performance of the three, failing while heating at about 1600 °C. There was a hold at about 1250 °C to adjust the flame and as the flame was subsequently moved towards the specimen, raising the temperature, it failed in the hot zone region due to damage and the force of the flame (Figure 12b). It was not surprising that the MI specimen would perform the worst since the matrix contains Si, which melts at ~1400 °C. It was interesting that failure occurred though, merely due to the force imposed on the center of the beam from the flame. We have subjected similar MI composites to similar temperatures with the lower velocity nozzle without failure for longer periods of time. This result, combined with the bending in the HNS-CVI specimen above, indicates that one needs to take precaution as to the amount of stress imposed by the flame and the temperature with respect to the creep/degradation mechanisms operating for a given composite. Thicker and wider specimens would be preferable for these types of tests.

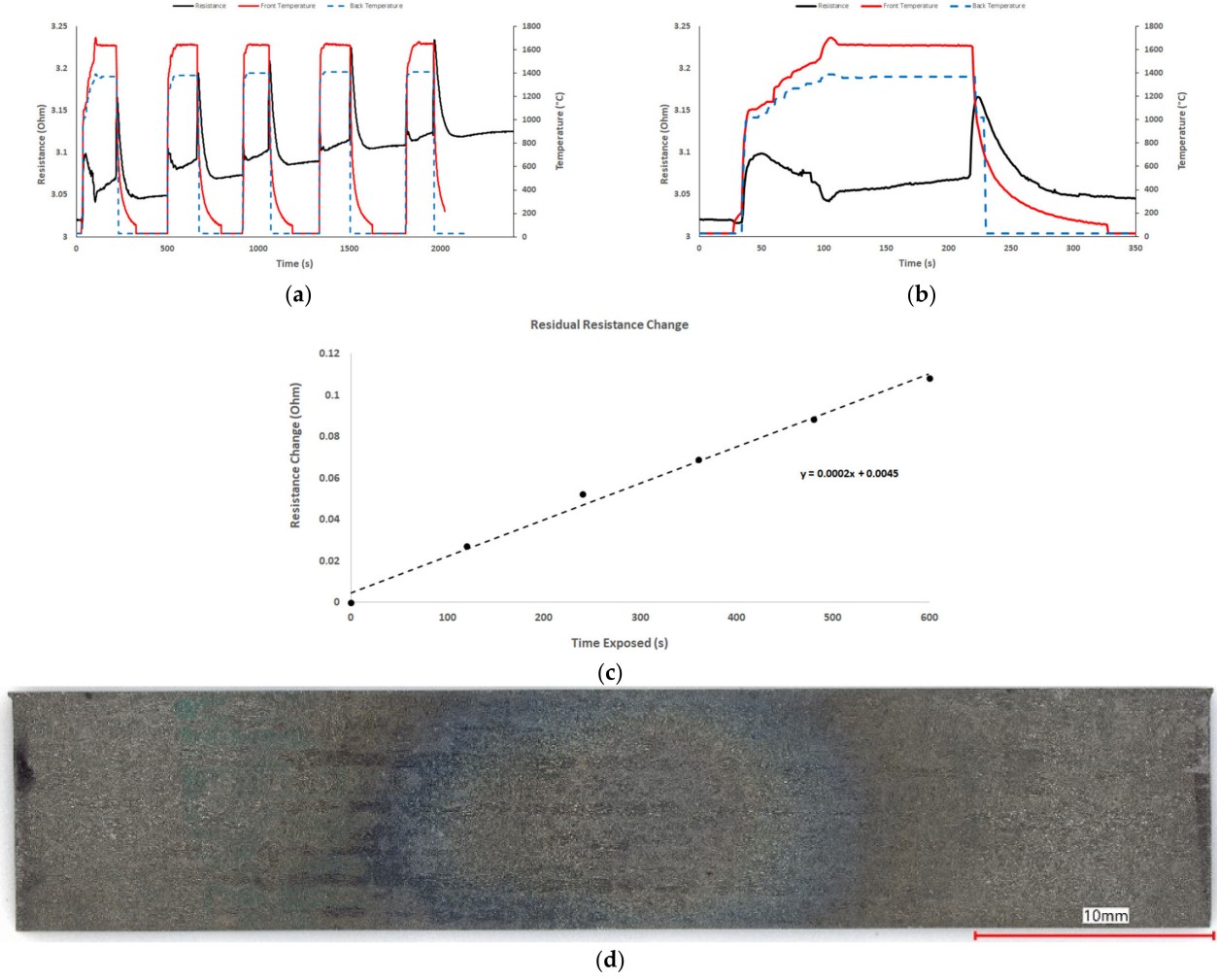

**Figure 11.** Experimental data acquired throughout the SA-Tyranno-Hex test (**a**) including front side and back side temperature as well as electrical resistance measured from the cold ends of the six inch specimen, (**b**) a magnification of the data in (**a**) for the first thermal-cycle, (**c**) the change in room temperature measured electrical resistance before and after each ther-mo-cycle iteration, and (**d**) the hot side image after testing.

It is evident that prior to failure, a significant amount of material was removed in the hot section of the flame. Figure 13 shows a higher magnification, 3D optical image with a depth profile near the center of the crater and fracture surface. The maximum depth of the crater was at least 0.38 mm, inferring that at least 25% of the material was removed due to the elevated temperature, high velocity flame condition.

At the very least, this type of testing provides a head-to-head type of test for prospective composites and/or other material/coating systems for a given condition. For example, given these three materials, the HNS-CVI composites and the SA-Tyranno-Hex performed the best, while the SA-MI was inadequate. Future work includes other composite and material/coating systems, tensile and/or compressive load conditions, other specimen configurations such as curved surfaces and other structures such as faceplate/core sandwich structures. Several issues need to be better understood and optimized, such as the force requirements that specimens can withstand given the velocity conditions. Additionally, the oxygen and water vapor content inherent in the combustion byproducts may not be representative of a given hypersonic environment—but it is more severe thermo-chemically. Some of the effects observed in this study are undoubtedly due to oxidation ingress into the interior of the composite from the exposed edges of the specimens, i.e., reaction of the environment with vulnerable interphase and open porosity regions at the cut edge. It would be advantageous to limit the edge effects by "sealing" the edges with an ad-

ditional matrix after machining. Nevertheless, the HVOF has proven to be an effective and simple approach to evaluating materials for these near hypersonic velocity, elevated temperature conditions.

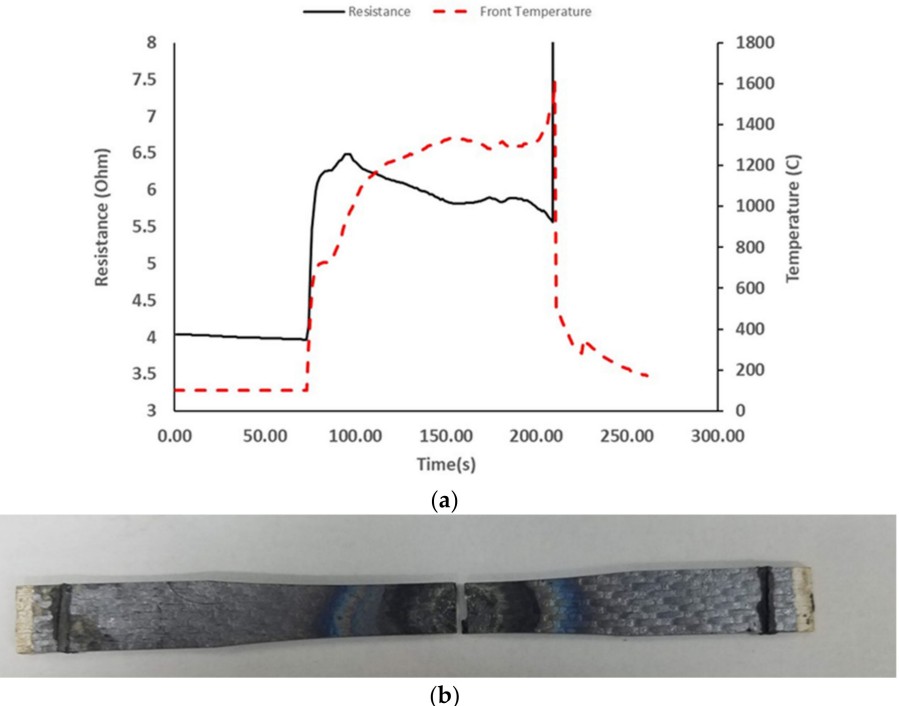

(a)

(b)

**Figure 12.** (**a**) Front side temperature and electrical resistance during heat up of SA-MI specimen and (**b**) optical micrograph of both halves of SA-MI specimen fractured while heating.

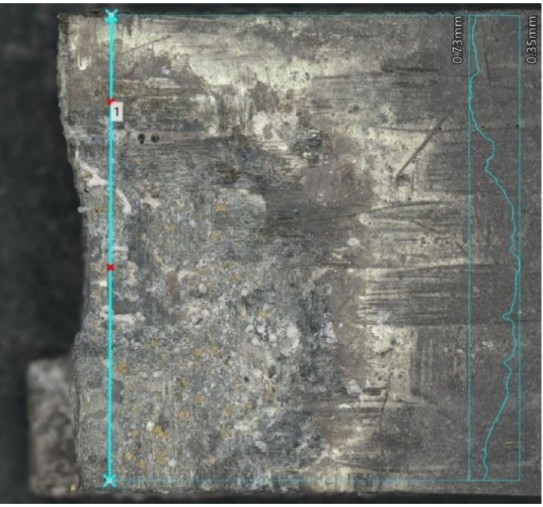

**Figure 13.** Optical image from a 3D optical microscope of face-view of fracture region of SA-MI specimen with depth profile across the width. The maximum depth ablated during high velocity exposure was about 0.38 mm.

## 4. Conclusions

The HVOF burner rig testing of ceramic matrix composites offers a new testing approach to assess the combined effects of temperature, mechanical loading and severe environment in a simple test rig. The fact that the flame is in the open allows for the incorporation of diagnostic measurement such as thermal imaging and health monitoring techniques. Electrical resistance is one example of the latter; however, other instrumentation

such as AE and DIC have been incorporated. With the added diagnostic equipment, more quantitative material responses and experimental conditions can be measured and used by modelers and designers to optimize thermo-chemical-mechanical models, assess design criteria and rank composite/coating performance.

The results of this study were qualitative. It is important to note that to understand and model the stressed-oxidation mechanisms at work will require tests that isolate the deleterious effects of stress, temperature, environment, velocity and/or thermal gradient. However, in practice, several if not all of these effects occur simultaneously, and this type of testing enables combinatorial effects to be understood that may not be discerned from narrower test conditions. In addition, as a "quick" test to down select materials for an application, this test would be ideal.

**Author Contributions:** Conceptualization, G.N.M., R.P.P. and L.H.; Investigation, R.P.P. and L.H.; Supervision, G.N.M.; Writing—original draft, G.N.M. All authors have read and agreed to the published version of the manuscript.

**Funding:** Part of this research (Case one) was funded by ONR grant number N00014-18-1-2646 (Dr. David Shiffler), Case two by a NAVAIR STTR contract number N68936-20-C-0018 (Dr. Sung Choi). Case three received some initial support from the same ONR grant as well as from a grant from Northrop Grumman Corporation.

**Institutional Review Board Statement:** Not applicable.

**Informed Consent Statement:** Not applicable.

**Conflicts of Interest:** The authors declare no conflict of interest.

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
