# Peer review of "The Versatility of HVOF Burner Rig Testing for Ceramic Matrix Composite Evaluation"

_jcs, doi:10.3390/jcs5080223_

Round 1

Reviewer 1 Report

The manuscript “the versatility of HVOF burner rig testing for ceramic matrix composite evaluation” studies the interest of HVOF facility to test materials and more particularly CMC under environments representative of combustion. This paper is interesting and shows the large possibilities offered by this kind of facility. Thus, HVOF burner rig can be used to study fatigue of materials under combustion conditions, to understand high temperature particle erosion and to assess materials under hypersonic conditions. The two first cases repeat previously published works and the third part is new. Thus the authors should improve this last part of the paper. Moreover, even if the paper does not deal with the materials, some information about the tested specimens is lacking.

Major corrections are required and listed:

Introduction :

P1, l22-23: the authors should replace “…hot section part applications” by “hot section parts”.

P1, l33 : the authors should replace “…can later bet tested…” by “..can later be tested…”

The authors should also mention the work of the Institute Pprime with MAATRE burner rig. Some papers of J. Cormier et al. are available in the literature.  Other facilities from DLR (high enthalpy wind tunnel) and Plasmatron (VKI) should also be mentionned.

Moreover some test facilities using laser instead of burner rig can also be cited in the introduction.

Experimental

Section 2.1

In this section, the authors should explain how the resistance is measured during the tests.

P3, l 93-96: The authors should explain why Chromium carbide is used to calculate velocity. This point is not clear.

In the case of the HVOF burner rig, the flame composition depends on the temperature. As H2O is present in the flame, this can largely influence SiC oxidation during the tests. Thus, the authors should add the flame composition in the table 1.

Section 2.2

The CMC specimens are too briefly described in table 1. The authors should detailed a little bit the materials (porosity level, fibre volume fraction, matrix composition,…) and thus the result part will be clearer. The thickness of the specimen can also be added in the table 1.

How many specimens are tested for each condition? As the microstructure of the specimens (porosity,…) can influence the results, the authors should link microstructure and the results. Moreover large amount of porosity is visible in figure 4, thus the relation between microstructure and properties is mandatory.

Results and discussions

P5, l144-155: the authors mention that thermal gradient, gas velocity and H2O content influence the lifetime of the specimen. Thus, direct comparison with fatigue test in a furnace is difficult as these three parameters are modified at the same time in the burner rig. The authors should highlight that it is the difficulty with this kind of experiments. Indeed, using HVOF burner rig, these three parameters cannot be dissociated and thus, it is hard to identify the main degradation mechanisms and the chronology of the degradation, which is useful for modeling and understanding.

Figure 9 a: the variations in the resistance are very small for this specimen compared to HNS-CVI specimen. How do the authors explain this? Are these variations really representative?

In the conclusions, the authors mention acoustic emission and digital image correlation at high temperature. The authors should include at least one example in the paper to complete the resistance measurement. These kind of instrumentations are really interesting and can help the analysis of these complex tests.

Author Response

The manuscript “the versatility of HVOF burner rig testing for ceramic matrix composite evaluation” studies the interest of HVOF facility to test materials and more particularly CMC under environments representative of combustion. This paper is interesting and shows the large possibilities offered by this kind of facility. Thus, HVOF burner rig can be used to study fatigue of materials under combustion conditions, to understand high temperature particle erosion and to assess materials under hypersonic conditions. The two first cases repeat previously published works and the third part is new. Thus the authors should improve this last part of the paper. Moreover, even if the paper does not deal with the materials, some information about the tested specimens is lacking.

Major corrections are required and listed:

Introduction :

P1, l22-23: the authors should replace “…hot section part applications” by “hot section parts”.

P1, l33 : the authors should replace “…can later bet tested…” by “..can later be tested…”

The authors should also mention the work of the Institute Pprime with MAATRE burner rig. Some papers of J. Cormier et al. are available in the literature.  Other facilities from DLR (high enthalpy wind tunnel) and Plasmatron (VKI) should also be mentionned.

Moreover some test facilities using laser instead of burner rig can also be cited in the introduction.

GRAMMATICAL ERRORS CORRECTED AND OTHER RIGS ADDED TO TABLE 1

Experimental

Section 2.1

In this section, the authors should explain how the resistance is measured during the tests.

P3, l 93-96: The authors should explain why Chromium carbide is used to calculate velocity. This point is not clear.

In the case of the HVOF burner rig, the flame composition depends on the temperature. As H2O is present in the flame, this can largely influence SiC oxidation during the tests. Thus, the authors should add the flame composition in the table 1.

RESISTANCE DESCRIPION ADDED WITH FIGURE. CHROMIUM CARBIDE EXPLAINED AND CLARIFIED: CHROME CARBIDE USED IN CASE 1 AND CASE 3 WHERE THE ILLUMINATION OF THE CARBIDE CAN BE PICKED UP BY HIGH SPEED CAMERA TO MEASURE VELOCITY. FOR EROSION THE ALUMINA PARTICLES THEMSELVES ARE SUFFICIENT TO MEASURE SPEED. ESTIMATED OXIDIZING SPECIES CONTENT IN THE FLAME ARE GIVEN IN THE TEXT OF THE FIRST PARAGRAPH OF 2.2 FOR THE TWO EXTREME CONDITIONS.

Section 2.2

The CMC specimens are too briefly described in table 1. The authors should detailed a little bit the materials (porosity level, fibre volume fraction, matrix composition,…) and thus the result part will be clearer. The thickness of the specimen can also be added in the table 1.

How many specimens are tested for each condition? As the microstructure of the specimens (porosity,…) can influence the results, the authors should link microstructure and the results. Moreover large amount of porosity is visible in figure 4, thus the relation between microstructure and properties is mandatory.

DETAILS WERE ADDED TO THE CMC SPECIMENS IN TABLE 2. EACH DATAPOINT CORRESPONDS TO A SPECIMEN. SOME MENTION OF POROSITY WAS MADE ESPECIALLY FOR THE HNS AND SA OF CASE 1 (BOTTOM OF PAGE 8) WITH RESPECT TO OXYGEN INGRES AND INCREASED PROPENSITY FOR INTERLAMINAR CRACKING. ADDITIONAL COMMENTS MADE THROUGHOUT HIGHLIGHTING THE DELETERIOUS EFFECT OF AS-MACHINED EDGES AND EXPOSED POROSITY AND INTERPHASES.

Results and discussions

P5, l144-155: the authors mention that thermal gradient, gas velocity and H2O content influence the lifetime of the specimen. Thus, direct comparison with fatigue test in a furnace is difficult as these three parameters are modified at the same time in the burner rig. The authors should highlight that it is the difficulty with this kind of experiments. Indeed, using HVOF burner rig, these three parameters cannot be dissociated and thus, it is hard to identify the main degradation mechanisms and the chronology of the degradation, which is useful for modeling and understanding.

Figure 9 a: the variations in the resistance are very small for this specimen compared to HNS-CVI specimen. How do the authors explain this? Are these variations really representative?

In the conclusions, the authors mention acoustic emission and digital image correlation at high temperature. The authors should include at least one example in the paper to complete the resistance measurement. These kind of instrumentations are really interesting and can help the analysis of these complex tests.

IN THE CONCLUSIONS, A PARAGRAPH WAS ADDED:

The results of this study were qualitative. It is important to note that to understand and model the stressed-oxidation mechanisms at work will require tests that isolate the deleterious effects of stress, temperature, environment, velocity and/or thermal gradient. However, in practice, several if not all of these effects occur simultaneously, and this type of testing enables combinatorial effects to be understood which may not be discerned from narrower test conditions. In addition, as a “quick” test to down select materials for an application, this test would be ideal.

REASONS FOR THE MILD CHANGES IN RESISTANCE ARE MENTIONED IN THE TEXT AT THE BOTOM OF PAGE 16:

Note that the changes in resistance for the Tyranno-Hex are slight compared to the HNS-CVI of Figure 8e. Two different specimens were tested of the SA Tyranno-Hex and both showed the same behavior. Since Tyranno-Hex consists essentially of SA fibers alone, the changes in resistance are related to the electrical properties of the fibers themselves and perhaps to the interconnected network of 0/90 fiber linkages formed during the hot pressing process. The HNS-CVI composites have two major current carriers, the HNS fibers and the CVI SiC. Certainly, oxidation of the interphase, carbonaceous species within the HNS fibers, the temperature dependence of the constituents themselves, damage due to matrix cracking will all complicate the temperature dependent electrical properties of the composite when compared to the Tyranno-Hex material.

Reviewer 2 Report

Despite very intersting works on the versality of the HVOF torch as a tool to test ceramic matrix composites under severe environments, the article is largely a compilation of several papers already published.

The results shoud focus on the unpublished third part. Several very interesting and appropriate diagnostic measurements, not shown in the article, are mentioned in the conclusions (thermal imaging correlation, health monitoring) it would be preferred to present these results in the core of the paper what would undeniably improve the content.

In the experimental method description, the conditions process (HVOF) could be presented with more details (torch diameter, total flow rate, flame stoechiometry, distance to the substrate to reach high velocity).

Some explanations could be given concerning the velocity measurement with chromia particule (particle size). In the erosion part, are you sure that the particle velocity is the same with different alumina particle size? 

In the third part, the observations after tensile test could advantangeously improve the understanding of the phenomena likely to cause a degradation of the properties.

Author Response

TABLE 3 WAS ADDED WHICH INCLUDES PRESSURES AND FLOW RATES FOR ALL THE TESTS. IN THE TEXT DISTANCE OF NOZZLE FROM FLAME AND FLAME HOT ZONE SIZE.

WE ADDED GREATER DETAIL TO CASE 3 INCLUDING THE FRACTURE SURFACE SHOWING CONSIDERABLE FIBER PULLOUT PRESUMABLY DUE TO DEGRADATION OF THE BN/FIBER INTERFACE.

RESISTANCE DESCRIPION ADDED WITH FIGURE. CHROMIUM CARBIDE EXPLAINED AND CLARIFIED: CHROME CARBIDE USED IN CASE 1 AND CASE 3 WHERE THE ILLUMINATION OF THE CARBIDE CAN BE PICKED UP BY HIGH SPEED CAMERA TO MEASURE VELOCITY. FOR EROSION THE ALUMINA PARTICLES THEMSELVES ARE SUFFICIENT TO MEASURE SPEED. ESTIMATED OXIDIZING SPECIES CONTENT IN THE FLAME ARE GIVEN IN THE TEXT OF THE FIRST PARAGRAPH OF 2.2 FOR THE TWO EXTREME CONDITIONS.

Reviewer 3 Report

The paper presented a lab-designed test platform to evaluate SiC-based ceramic matrix composites (CMCs) under several extreme thermo-mechanical conditions. The research is comprehensive and the paper is well organized. The paper will be helpful and interested in the field of SiC-based CMCs served in harsh environments. I would like to recommend the publication of this paper. 

Author Response

No reply

Round 2

Reviewer 1 Report

The authors have taken into account all the comments of the reviewer. The paper is more complete and readable and can be accepted for publication.

Reviewer 2 Report

Thanks for having taken my comments into account

Author Response

The authors have correctly addressed the major points of concern. The paper is now almost OK for publication.

A couple of improvements can be made :

Table 1 : No BRICS testing facilities are cited, whereas they are numerous. I recommend selecting at least 3 to 4 significant papers from institutions like

NorthWestern Polytechnical University, NWPU Xi'An, China

Beijing University of Aeronautics & Astronautics, BUAA Beijing, China

Shanghai University, Shanghai, China

Central South University, CSU, Changsha, China

ITA, S. José dos Campos, Brazil

National University of Science and Technology “MISiS”, Moscow, Russia

The associated publications should appear in the reference list.

WE COULD NOT FIND REFERENCES TO BRICS TEST FACILITY. COULD THE REVIEWER PROVIDE SOME?

#129-131 : Please correct the following sentence :

"For Case 1 and 3, velocity was calculated by spraying a small amount of chrome carbide initially at the HVOF conditions for a given experiment particles via the powder feeder."

into :

"For Cases 1 and 3, velocity was calculated by spraying a small amount of chromium carbide particles, initially at the HVOF conditions for a given experiment, via the powder feeder."

  • CHANGED

#133: Please provide a web page pointing to the "Tracker" open software.

  • FOOTNOTE ADDED
